Corrected: Author Correction

# A practical guide for mutational signature analysis in hematological malignancies

Francesco Maura[1,2,3], Andrea Degasperi [3,4,5], Ferran Nadeu [6,7], Daniel Leongamornlert[3], Helen Davies[3,4,5], Luiza Moore [3], Romina Royo[8], Bachisio Ziccheddu[9], Xose S. Puente [10,11], Herve Avet-Loiseau[12], Peter J. Campbell[3], Serena Nik-Zainal[3,4,5], Elias Campo[6,7,8], Nikhil Munshi[13,14] & Niccolò Bolli[2,9]

Analysis of mutational signatures is becoming routine in cancer genomics, with implications for pathogenesis, classification, prognosis, and even treatment decisions. However, the field lacks a consensus on analysis and result interpretation. Using whole-genome sequencing of multiple myeloma (MM), chronic lymphocytic leukemia (CLL) and acute myeloid leukemia, we compare the performance of public signature analysis tools. We describe caveats and pitfalls of de novo signature extraction and fitting approaches, reporting on common inaccuracies: erroneous signature assignment, identification of localized hyper-mutational processes, overcalling of signatures. We provide reproducible solutions to solve these issues and use orthogonal approaches to validate our results. We show how a comprehensive mutational signature analysis may provide relevant biological insights, reporting evidence of c-AID activity among unmutated CLL cases or the absence of BRCA1/BRCA2-mediated homologous recombination deficiency in a MM cohort. Finally, we propose a general analysis framework to ensure production of accurate and reproducible mutational signature data.

[1] Myeloma Service, Department of Medicine, Memorial Sloan Kettering Cancer Center, New York 10065 NY, USA. [2] Department of Oncology and Hemato-Oncology, University of Milan, Via Festa del Perdono 7, Milan 20122, Italy. [3] Cancer, Ageing, and Somatic Mutation Programme, Wellcome Sanger Institute, Hinxton, Cambridgeshire CB10 1SA, UK. [4] Department of Medical Genetics, Cambridge University Hospitals NHS Foundation Trust, Cambridge CB2 0QQ, UK. [5] MRC Cancer Unit, University of Cambridge, Hutchison/MRC Research Centre, Cambridge Biomedical Campus, Cambridge CB2 0XZ, UK. [6] Patologia Molecular de Neoplàsies Limfoides, Institut d'Investigacions Biomèdiques August Pi i Sunyer (IDIBAPS), 08036 Barcelona, Spain. [7] Centro de Investigación Biomédica en Red de Cáncer (CIBERONC), 28029 Madrid, Spain. [8] Barcelona Supercomputing Center (BSC), Joint BSC-CRG-IRB Research Program in Computational Biology, 08036 Barcelona, Spain. [9] Department of Clinical Oncology and Hematology, Fondazione IRCCS Istituto Nazionale dei Tumori, Milan 20133, Italy. [10] Unitat Hematopatologia, Hospital Clínic of Barcelona, Universitat de Barcelona, 08036 Barcelona, Spain. [11] Departamento de Bioquimica y Biologia Molecular, Instituto Universitario de Oncologia (IUOPA), Universidad de Oviedo, Oviedo 33003, Spain. [12] IUC-Oncopole, and CRCT INSERM U1037, 31100 Toulouse, France. [13] Jerome Lipper Multiple Myeloma Center, Dana–Farber Cancer Institute, Harvard Medical School, Boston 02215 MA, USA. [14] Veterans Administration Boston Healthcare System, West Roxbury 02130 MA, USA. Correspondence and requests for materials should be addressed to F. M. (email: mauraf@mskcc.org) or to N.B. (email: niccolo.bolli@unimi.it)

The advent of next generation sequencing has profoundly changed both the research and clinical approach to cancer in the last 10 years[1]. While the cancer genome landscape may be composed of thousands of events, only a minimal fraction of them can be considered as drivers[2–5]. Despite the majority of tumor mutations do not have a functional role, the entire coding and non-coding mutational catalog can be extremely informative for the identification of the mutational processes operative in different cancer types during initiation and progression[4,6–10].

Historically, a simple analysis of single-nucleotide variants (SNVs) as a six-class mutational spectrum (C·G → A·T, C·G → G·C, C·G → T·A, T·A → A·T, T·A → C·G, and T·A → G·C) has highlighted how different cancer types are characterized by different contributions from each class, some of which strongly associated with distinct exogenous carcinogens exposure[11,12]. For example, the C·G → A·T transversion is related to smoking in lung cancer samples[13], and the C·G → T·A transition is significantly over-represented in skin cancers related to UV light exposure[11,12,14]. Following on from these preliminary observations, different approaches have been suggested to gain resolution in the analysis of these so called mutational signatures. Combining the six possible SNV classes together with their trinucleotide contexts (i.e., the bases 5′ and 3′ of the mutated nucleotide) all SNVs have been classified into 96 possible combinations[6,7,15]. This classification has then been used to extract >30 different mutational signatures with a non-negative matrix factorization (NNMF) approach from a large series of whole-genome (WGS) and exome (WES) sequencing data[6,16,17]. Some of these signatures are specifically associated with defects of DNA repair mechanisms, exposure to exogenous carcinogens, or different patterns of structural variants (SVs), suggesting they truly reflect known and unknown mutational processes shaping the genome of each cancer type[10,15,17–20]. Further to corroborating their biological relevance, some mutational signatures are also associated with a distinct clinical outcome and emerged as potential biomarkers for novel target therapies[18,19,21,22].

Since this initial effort, several alternative approaches to NNMF have been proposed to improve the mathematical efficacy and biological accuracy of mutational signatures extraction from the 96-class profile of each cancer[6,7,10,23–29]. However, the field of mutational signature extraction still lacks a unanimous consensus and standardization of analysis, often resulting in discrepancies between results from similar datasets obtained using different methodological approaches[4,9,10,21,22,30–33]. As WGS and WES are becoming common practice, with implications for both basic and translational research, we believe that more should be done to improve the performance and the reproducibility of mutational signature analysis.

In this study, we use different publicly available bioinformatics tools to analyze public datasets from multiple myeloma (MM) and chronic lymphocytic leukemia (CLL) samples, and validate our findings in additional published and unpublished sequencing data from acute myeloid leukemia (AML) samples, to summarize the main factors that should be considered in a high-confidence mutational signature analysis. We discuss sources of bias and pitfalls, and provide a rational and practical approach that could be validated in other independent studies.

## Results

### Common issues of mutational signature analysis. All different mutational signature analysis algorithms produce a decomposition matrix $C \approx SE$, where $C$ is the catalog matrix, with mutation types as rows and samples as columns, $S$ is the signature matrix, with mutation types as rows and signatures as columns, and $E$ is the exposure matrix, with signatures as rows and samples as columns (Supplementary Fig. 1). Nevertheless, different approaches can be divided in two main groups: (i) the ones that allow de novo signature extraction (e.g., the NNMF framework from Alexandrov et al.)[6], where given a matrix $C$ the algorithm finds matrices $S$ and $E$ such that $C \approx SE$, and (ii) the ones that fit the 96-mutational catalog to a pre-selected list of signatures (e.g., the 30 COSMIC signatures), where given $C$ and $S$ the algorithm finds $E$ such that $C \approx SE$. An example of algorithm of the second group is deconstructSigs[24]. Both approaches can be extremely informative in different settings, though it is not always easy to determine when and how to use one or the other. Working with mutational signatures analysis with either group of algorithms, we identified three main issues. The first is the ambiguous signature assignment that occurs when different combinations of signatures can explain equally well the same mutational catalog. This issue may arise when multiple so called flat mutational signatures are potentially present in the same data set (e.g., COSMIC signatures 3, 5, and 8) (Supplementary Fig. 2)[6,31,34]. The second usually occurs when localized mutational processes are not investigated. In fact, when a signature extraction is performed using all the mutations found in a genome (or exome), only mutational signatures induced by mutational processes that act across the entire genome are usually identified. Localized mutational processes are often responsible for a small proportion of the total number of genome-wide mutations, and thus are generally missed[9,10,35,36]. The third common issue is the bleeding of signatures. It is biologically sound to assume that each cancer sample presents the activity of a limited number of mutational processes. If an extraction is performed on a heterogeneous set of samples, it is possible that signatures present in only part of the set are also erroneously assigned to the entire set. This is mostly due to the algorithms' assumption that all analyzed samples share a similar mutational signature landscape and to the fact that some signatures are similar to each other.

### Mutational signature extraction vs. fitting. As mentioned above, a signature analysis can be performed using either a de novo extraction or a fitting approach based on a pre-selected reference list of known signatures (e.g., the 30 COSMIC signatures).

The first approach extracts recurrent patterns of variants in their trinucleotide context from the input data allowing the unbiased identification of both known and novel mutational processes. However, the weakness of this approach is that extracted signatures often do not appear identical to the reference ones. Common problems are: (i) union of co-occurrent multiple signatures into one; (ii) over splitting of one mutational signature into two or more. All these factors can significantly impact the assignment of extracted signatures to the reference ones[6,31], and this may introduce bias in the estimation of each signature's activity in the samples.

The second approach fits the input data to a suitable reference list of mutational signatures, allowing a better estimation of each signature's relative and absolute contribution for each sample. However, a fitting approach is not able to discover any novel signature and thus needs a priori knowledge of which mutational processes may be operative in that sample cohort. Furthermore, these approaches may be prone to overfitting leading to signature bleeding, i.e., they may assign all signatures from the reference list to all samples. Therefore, before running any fitting algorithm, it is crucial to have at least some knowledge about which mutational processes are operative in the samples to avoid both false positives (overfitting of signatures) and false negatives (missing novel mutational process).

To provide an example of the problems that a fitting algorithm can pose to the interpretation of data if analyzed without any a

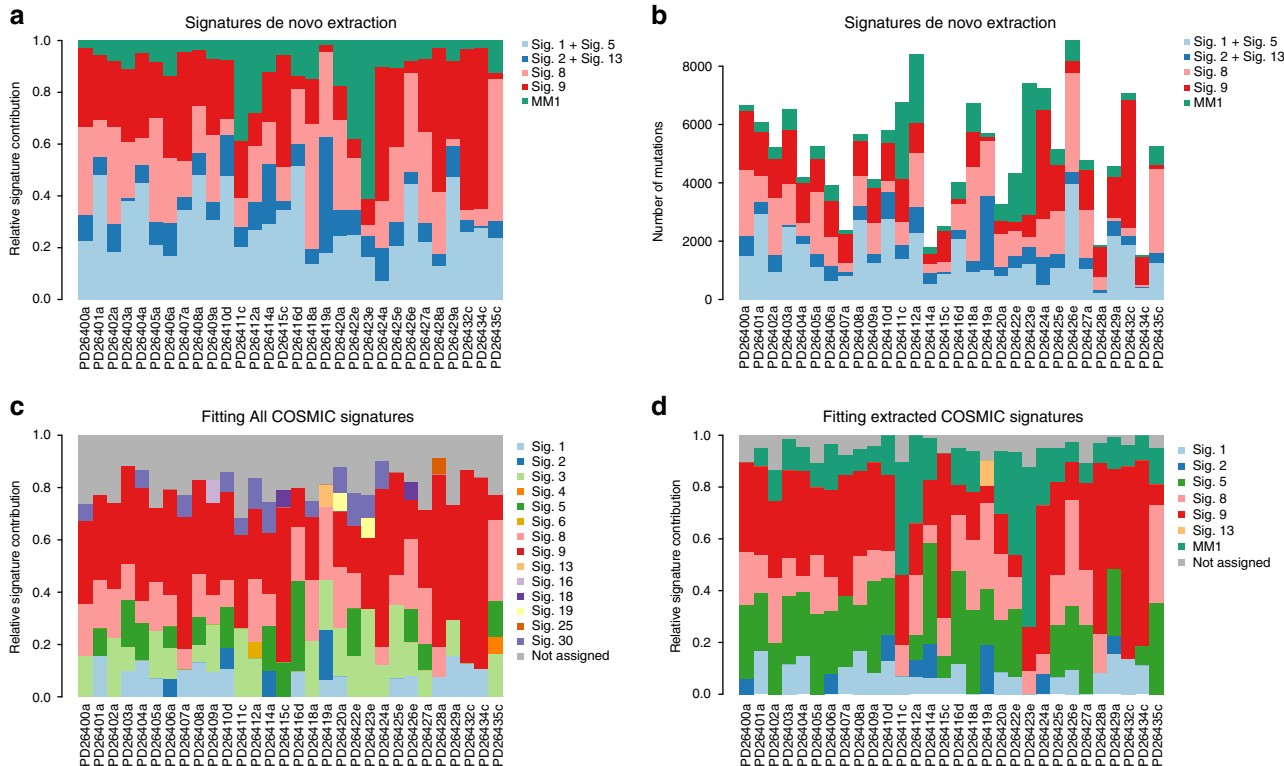

**Fig. 1** Mutational signature de novo extraction vs. fitting. **a**, **b** The Alexandrov et al. NNMF framework[6,7]. From the 96-mutational classes, NNMF extracted the signatures' relative (**a**) and absolute (**b**) contribution among 30 MMs. **c** Running deconstructSig including all 30 COSMIC signatures several mutational processes were forced to be extracted (i.e., Signature 4). Furthermore, the new mutational process MM1 was not detected, being not included in the 30 COSMIC signatures. **d** Conversely, running the same fitting approach based on the mutational signature catalog extracted by NNMF, each signature contribution was better estimated for each patient. Sig. = signature

priori knowledge, we used a cohort of 30 MM cases (Supplementary Table 1), which have been extensively characterized from a genomic point of view. Here, we first applied NNMF-based, de novo extraction algorithms, i.e., the framework from Alexandrov et al.[6,7]. (Fig. 1a, b) and the NNMF approach of the mutationalPatterns R package[37] (Supplementary Sofware 1). Both NNMF approaches extracted five signatures: the clock-like signatures (Signature 1 and 5 merged together), APOBEC (Signature 2), Signature 8, Signature 9, and a new signature named MM1, again highlighting the impact that NNMF approaches can have in new signature discovery (Supplementary Data 1)[6,9,16,23]. Then, using the same input data we then ran two fitting approaches (deconstructSigs and the fitting approach of mutationalPatterns) without a priori knowledge of the active mutational processes in MM and therefore including all 30 COSMIC signatures. DeconstructSigs forced the extraction of a large number of signatures, including ones not previously extracted by NNMF, and some of which clearly representing false positives (Fig. 1c and Supplementary Sofware 1). For example, the contribution of tobacco-smoking (COSMIC Signature 4) to MM development can most likely be ruled out, as can the contribution of the liver-specific Signature 16 (Fig. 1c)[17,31,38]. Furthermore, the new signature MM1 was not identified, simply because it was not included in the COSMIC catalog. To reduce false positives, some corrections can be applied to the fitting approach. For example, deconstructSigs uses forward selection to estimate a minimal number of signatures, and removes a signature's contribution to a sample if it accounts for <6% of the sample's mutations. In contrast, mutationalPatterns fitting approach does not introduce any correction while attempting to fit all 30 COSMIC signatures. In this case, a false-positive

minimal contribution of unlikely signatures was detected in all patients (mismatch repair, UV light, tobacco-smoking etc.) (Supplementary Sofware 1). Altogether, this shows that fitting approaches may crucially alter the inferred mutational signature landscape in MM. Conversely, when we ran deconstructSigs and mutationalPatterns imputing the shortlist of COSMIC signatures previously identified by the extraction approaches (i.e., NNMF), this led to a more biologically sound assignment and quantification of the absolute and relative contribution of each process (including the new signature MM1) for each sample, significantly reducing the false-positive signatures (Fig. 1d and Supplementary Sofware 1).

**Absence of BRCA-mediated Homologous Recombination Deficiency in MM.** The genomic profile of MM is characterized by several recurrent and private cytogenetic aberrations, making it one of the most complex hematological malignancies from this point of view[3,21,39–44]. Recently, using a fitting approach like deconstructSigs with default parameters[24], a potential activity from Signature 3 has been proposed in a significant fraction of MMs[32]. This mutational signature is well-known to correlate with BRCA1 and BRCA2 bi-allelic loss and homologous repair deficiency (HRD) in different solid cancers[6,18,20,45]. Signature 3 was indeed observed in our MMs when either mutationalPatterns or deconstructSig fitting approaches were run using all 30 COMISC signatures (Fig. 1c and Supplementary Sofware 1), but not observed in our signature extraction.

To positively confirm whether or not signature 3 is present in our samples, we used two validation strategies: (1) determine whether the pattern of Signature 3 is necessary to explain the

mutational patterns observed in the samples; (2) analyze additional genomic features to determine the presence of HRD.

First, to establish whether Signature 3 is required to explain the catalog of mutational signatures in our samples, we determined whether including or not Signature 3 in our analysis would affect the reconstruction error, i.e., the difference between the original catalogs and the fitted linear combination of signatures for each sample (see Methods). The inclusion of Signature 3 produced a statistically significant lower reconstruction error (measured as KL divergence, root mean squared error (RMSE) or cosine similarities), which can be attributed to the inclusion of an additional signature in the linear combination. However, the reconstruction error is not qualitatively different in the absence of Signature 3 (Supplementary Fig. 3a–c, g–i). In contrast, when Signature 3 is used in place of either Signature 8 or Signature 5, we have a qualitative increase in the reconstruction error (Supplementary Fig. 3d–f, j–l). Interestingly, when Signature 3 is excluded, the mutations that were assigned to Signature 3 seem to be reassigned mostly to the other flat Signatures 8 and 5 (Supplementary Fig. 4). This evidence indicates that Signature 3 is not necessary to explain the patterns of SNV mutations in the samples. Conversely, Signature 8 and Signature 5 emerged as the most significant processes, and the ones that are likely active.

Next, we used an orthogonal approach to detect the presence of BRCA1/BRCA2-like HRD in our MM samples (Fig. 2): to this end, we applied the recently published HRDetect tool[18], a highly accurate classifier that estimates the presence of BRCA1/BRCA2-like HRD in solid cancers, trained on multiple mutational patterns, including COSMIC Signature 3, COSMIC Signature 8, microhomology-mediated deletions, Rearrangement Signatures 3 and 5 (unclustered short tandem duplications and deletions, respectively)[20] and the HRD index[46]. If we exclude Signature 3 from our analysis, none of the 30 MM samples would be classified as HRD, as they do not appear to be enriched with the patterns that are typical of the BRCA1/BRCA2-type of HRD: there is a low proportion of microhomology-mediated type of small deletions, the HRD-LOH index[46] is low, and there is a limited number of 1–100 Kb deletions (Rearrangement Signature 5) and 1–100 Kb tandem duplications (Rearrangement Signature 3) (Fig. 2a, Supplementary Figs. 5 and 6). After including both Signature 3 and Signature 8, only one sample (PD26419a) would show an elevated HRDetect score (Fig. 2b). This sample, characterized by multiple complex events and chromothripsis[47], is likely to be a false positive generated by the erroneous inclusion of Signature 3 in our analysis. In fact, it lacked the characteristic unclustered genome-wide rearrangements and predominance of microhomology-mediated type of small deletions (Fig. 3a, b and Supplementary Figs. 5 and 6). Finally, if we included Signature 3, we would expect some correlation between the HRDetect score and the assignment of Signature 3, since they both correlate with HRD. However, such correlation is absent in our analysis (Fig. 2b, c).

In conclusion, fitting approaches like deconstructSigs (or mutational pattern) tend to force the assignment of flat signatures, such as Signature 3, to samples when all 30 COSMIC signatures are used as input (Fig. 1c, Fig. 3a, and Supplementary Sofware 1). However, we demonstrated that Signature 3 is not necessary to explain the mutational patterns of MM samples, which furthermore do not show a genomic landscape consistent with BRCA1/BRCA2 loss and its related HRD in terms of 96-class profiles, number of microhomology-mediated deletions and internal tandem duplications as compared to breast cancer (Fig. 3b, c and Supplementary Figs. 5 and 6). We therefore suggest that Signature 3 (and consequently BRCA1/2-mediated HRD) is not biologically active in our MM samples, and it likely represents a false-positive call. Rather, we believe that the right signatures to be annotated in these samples are Signature 8,

widely involved in solid and hematological cancers with an unknown etiology[6], and Signature 5, a flat clock-like process present in normal and cancer tissues[16]. This of course does not exclude the possibility that a larger cohort of MM samples may show cases of BRCA1/2-like HRD, though again, we have no evidence that this is the case in our cohort.

**Localized hypermutation.** When a naive B-cell passes through the germinal center (GC), it is usually exposed to the activity of activation-induced cytidine deaminase (AID), which is responsible for a very unique genetic process called somatic hypermutation (SHM) of the B-cell receptor (BCR) variable region (VDJ)[48]. This mutational process plays a critical role in the antibody diversification promoting mutations and aminoacidic changes on immunoglobulin heavy and light chain (IGH/IGK/IGL) genes in order to increase the B-cell receptor (BCR) affinity to distinct antigens[48]. Chronic lymphocytic leukemia (CLL) is well-known to be characterized by two main biological subgroups: one dependent on GC exposure and one independent (Supplementary Data 2). These are differentially diagnosed by recognizing patterns of AID-driven somatic hypermutation in one group (mutated CLL, M-CLL) and not in the other (unmutated CLL, U-CLL)[5,49–53]. MM and M-CLL are post-GC lymphoproliferative malignancies, and their (pre)malignant cells are exposed to AID activity[9,32]. This mutational process, named canonical-AID (c-AID), has been known for years and is specifically active on IGH/IGK/IGL loci[48,54,55]; however, thanks to mutational signatures analysis, an alternative AID-driven mutational process has been recently observed genome-wide in all post-GC lymphoproliferative disorders[6,10,52,53]. This process was named non-canonical AID (nc-AID; COSMIC Signature 9) and differs from the above-mentioned c-AID in terms of preferential trinucleotide context, genomic distribution and associated cell cycle phase (Supplementary Fig. 7)[55]. In contrast to nc-AID, the c-AID signature is generally not identified by de novo signature extraction algorithms because it is localized and its limited activity is diluted below the threshold of detection by the larger number of genome-wide mutations generated by other processes (see the lack of its detection in all MM and CLL samples in Fig. 1, Supplementary Data 1, and Supplementary Sofware 1 and 2)[9,10,52]. However, identification of the mutational burden of c-AID and its aberrant targets (e.g., BCL6[54]) can be extremely informative to compare the genomic landscape of different lymphoproliferative disorders and their different biological origins. The characterization of this localized mutational process can be performed in two ways, with either extraction or fitting algorithms after inclusion of the c-AID 96-class profile (Supplementary Fig. 7), currently not part of the COSMIC panel: (1) Considering only hypermutated regions, i.e., those with >5 mutations with a median inter-mutational distance of < 1 Kb;[6,9,15,47] (2) Considering only mutations that occur within known c-AID targets, in particular the IGH/IGK/IGL loci[52]. Both approaches can identify c-AID in both MMs and CLLs (Fig. 4), i.e., two neoplasms where activity of this enzyme is expected. Interestingly, and confirming other previous preliminary data[10], c-AID activity was also detected in a fraction of U-CLL patients despite the GC-independent pathogenesis. Specifically, in MM and to a greater extent in M-CLL, >10% of these mutations were observed within coding genes, in particular across the VDJ region of the IGH locus; conversely, among U-CLL this activity involved mostly the non-coding part of the IGH locus, in particular within the class switch recombination loci (Supplementary Fig. 8a–d). These data are in line with the ability of WES to identify c-AID signature within the IG loci only among M-CLL cases[52], and strengthen the need for WGS for a comprehensive signature analysis.

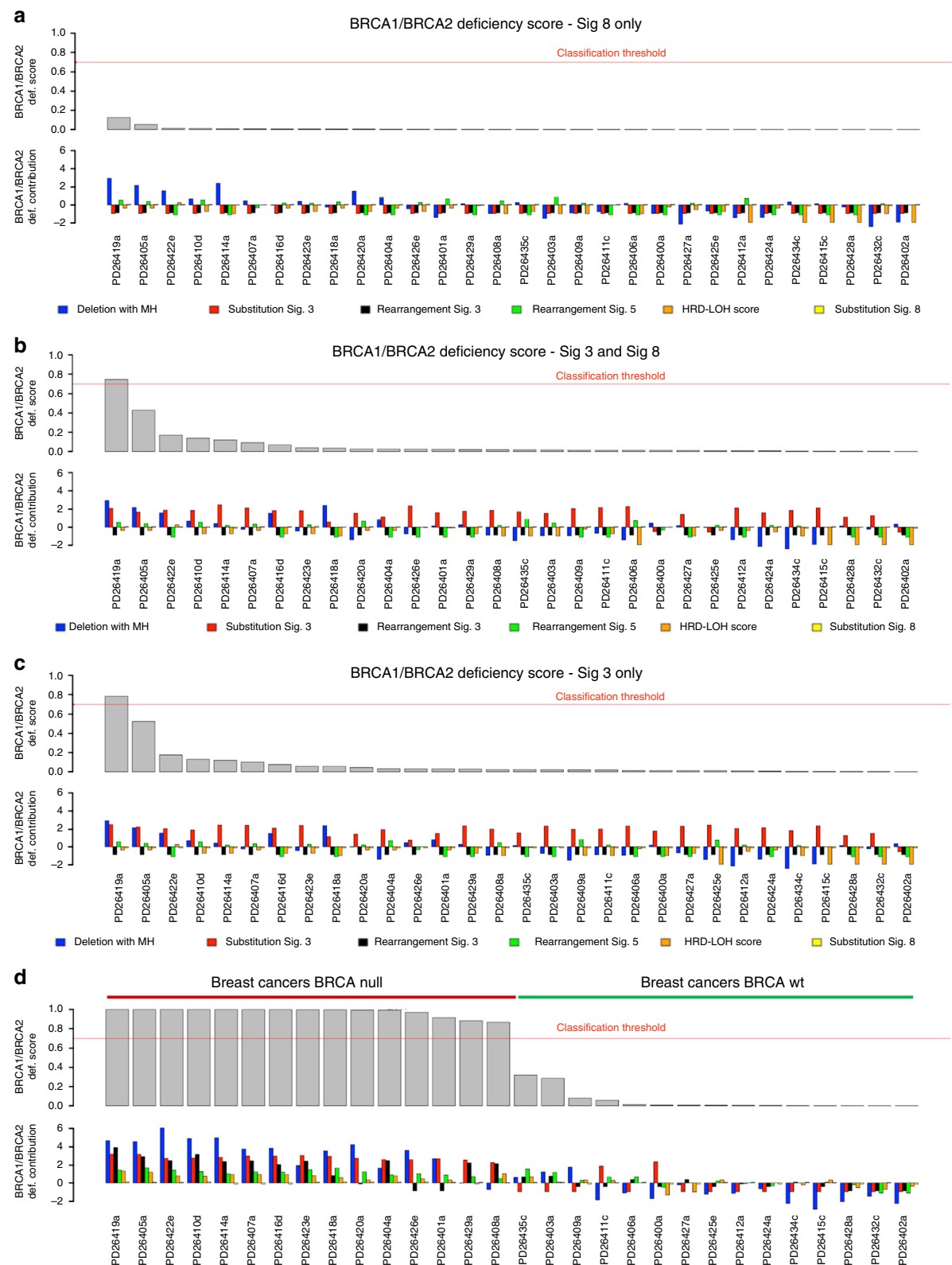

**Fig. 2** HRDetect BRCA1/BRCA2 deficiency scores in MM. HRDetect was used to analyze the BRCA1/BRCA2 deficiency scores in MM samples **a** including only signature 8, **b** including both signatures 3 and 8, and **c** including only signature 3. In **d**, the same analysis was performed in 15 BRCA null and 15 BRCA wt breast cancers[18]. Scores are ordered from highest to lowest and a classification threshold of 0.7 is used to classify samples as HRD-positive (see Davies et al.[18]). Below each score, the contribution of the six features that are used by HRDetect is shown. Each contribution is given by the amount of a feature in a sample, log-transformed and standardized according to mean and standard deviation of the features in Davies et al.[18] and finally multiplied by the corresponding HRDetect logistic regression coefficient. Thus, a positive contribution indicates a feature value higher than the average of the HRDetect original training set, and feature contributions are directly comparable. Sig. = signature

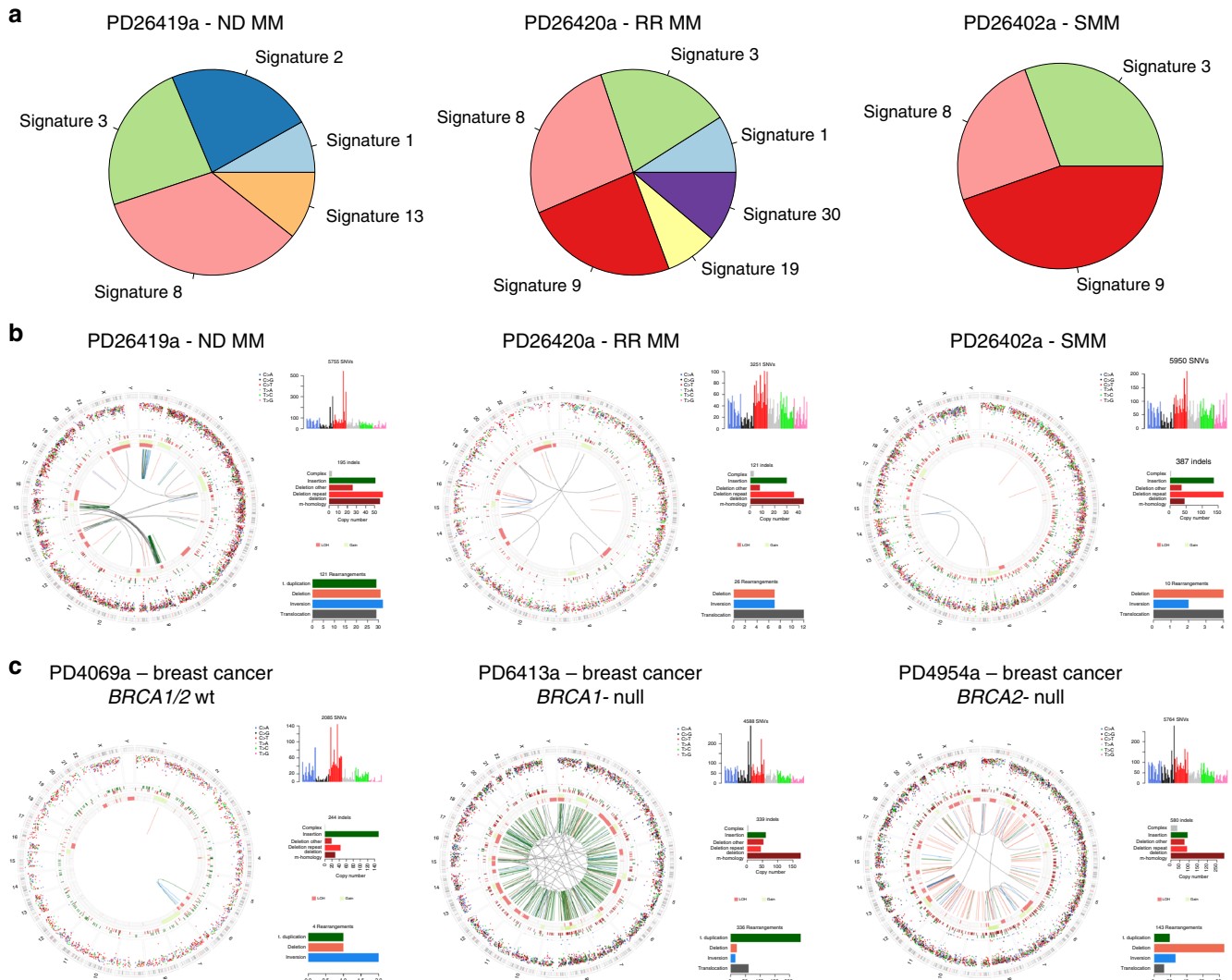

**Fig. 3** Absence of BRCA-driven HRD in MM. **a** Pie charts showing the relative signature composition according to DeconstructSig in three MM cases, without a prior knowledge of which signatures are involved or detected by NNMF. Testing all 30 COSMIC mutational signatures, Signature 3 is extracted is all samples. **b** Circos plot of three MMs (ND = newly diagnosed; RR = relapsed/refractory; SMM = smoldering MM) where deconstructSig extracted a significant Signature 3 contribution. From the external ring to the internal: mutations, (vertically plotted according to their inter-mutational distance and where the color of each dot represents the mutation class), indels (dark green = insertion; and brown = deletion); copy number variants (red = deletions, green = gain), rearrangements (blue = inversion, red = deletions, green = ITD, black = translocations). PD26419a is the only patient with a slightly high HRDetect score when analyzed including Signature 3. **c** Circos plots of a breast cancer sample without BRCA deficiency (PD4069a), one with BRCA1 deficiency (PD6413a) and one with BRCA2 deficiency (PD4954a). The MM genomic landscape shows significant differences to the two BRCA-deficient breast cancers, in particular in terms of numbers of indels and SVs, suggesting BRCA-driven HRD is not present in the MM samples analyzed

Furthermore, in contrast to MM and M-CLL cases, nc-AID was not active in IGH regions from U-CLL cases (Fig. 4). Confirming previous reports on a potential ongoing AID activity in U-CLLs[10], a significant higher fraction of subclonal c-AID mutations (i.e., late mutations) was observed among this group of CLLs (Supplementary Fig. 8e). Conversely, c-AID mutations were mostly detected at clonal level (i.e., early mutations) in M-CLL and MM, confirming the recently reported decreased AID activity in late stages of these diseases[9,10]. Overall, these data suggest a possible non-VDJ and GC-independent role of c-AID among U-CLLs (Fig. 4)[10,56].

To better characterize the c-AID activity on known loci, we usually prefer to focus on mutations within known c-AID targets rather than to identify hypermutated regions. In fact, most of c-AID mutations occurred close to different VDJ breakpoints, where distant genomic regions are joined by the RAG/AID complex during early stage of B-cell development before the GC exposure[48].

This means that inter-mutational genomic distance does not reflect the true position of these mutations and should be corrected for the VDJ structure to identify mutations caused by c-AID activity (Supplementary Fig. 9). This also applies to localized hypermutation events (i.e., kataegis) around complex structural variants (i.e., chromothripsis), where the cancer chromosomal structure significantly differs from the reference[15,47].

As mentioned above, this kind of analysis can be also directed on known c-AID aberrant targets, such as *BCL6*, allowing the characterization of clustered mutational processes active around these critical oncogenes and key GC regulators (Supplementary Fig. 10)[54]. In our series, *BCL6* was involved in localized mutational processes in M-CLL and MM reflecting their GC exposure, as expected; conversely, U-CLLs did not show any evidence of this process, confirming the GC-independent pathogenesis and suggesting the existence of a GC-unrelated AID activity in this group of patients.

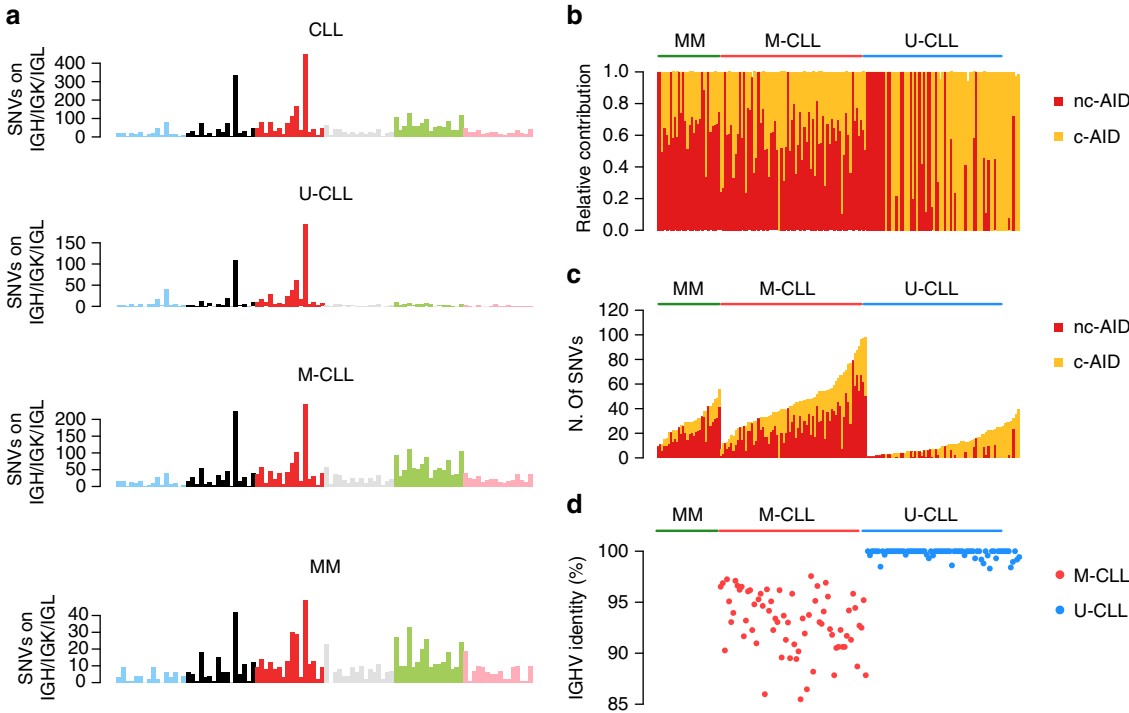

**Fig. 4** Mutational signature landscape of immunoglobulin loci. **a** The 96-mutational classes of all SNV within IGH/IGK/IGL loci. Canonical AID (c-AID) represented the main mutational process within these regions in all tested hematological malignancies, including U-CLLs as recently described[3,10,52]. **b**, **c** Mutational signature relative (**b**) and absolute (**c**) contribution within IGH/IGK/IGL loci for each sample tested by deconstructSig. **d** The Sanger-sequencing-based IGHV mutational status available for each CLL case. Sig. = signature

SHM is only present in post-GC B-cells, however it is not the only example of localized hypermutation in cancer. An instance of localized hypermutation termed kataegis has been found across many cancer types and is often promoted by aberrant activity of the APOBEC family of DNA deaminases[47,57]. We have previously reported widespread and localized activity of APOBEC in MM (Fig. 5a–c)[9] where it is recurrently associated with complex rearrangements such as chromothripsis, similarly to what has been reported in several other solid cancers[47]. Furthermore, here we report the first case of APOBEC-mediated kataegis in a therapy-related AML case, again associated with a complex rearrangement (Fig. 5d–f). Previously, APOBEC was never reported as active in AML[6,31]. Overall, our findings stress the importance of performing ad-hoc signature analysis in localized mutational events, since this can highlight specific pathogenetic mechanisms across different cancer types.

**Inter-sample bleeding.** Both WGS and WES data have clearly shown that M-CLL samples are characterized by a very distinct mutational process (COSMIC Signature 9), reflective of the genome-wide nc-AID activity within the GC[6,10,52]. Conversely, we would expect the absence of nc-AID signature in U-CLL, as these cases do not develop through the GC. To validate this assumption, we performed a de novo signature extraction on all CLLs, using either the Alexandrov et al.[6] framework or the mutationalPatterns[37] NNMF function (Supplementary Data 1). A nc-AID signature was assigned to all samples, with high activity in M-CLL samples and a much lower contribution in U-CLLs (Fig. 6 and Supplementary Sofware 2). This represents a typical example of inter-sample bleeding effect caused by the assumption that all these samples shared a similar mutational landscape. This incorrect assignment would not be readily highlighted if the biology underlying CLL pathogenesis was not thoroughly known. To obviate this problem, we propose two approaches. In the first,

we re-fit the extracted signatures. Here, signatures are first extracted with a de novo approach. Then, a fitting algorithm such as deconstructSigs is applied using only the signatures extracted by NNMF to clean up low-contribution signatures, mostly representing false positives (Fig. 6b, c). The second approach involves performing separate extractions. NNMF is run independently on two sets of samples, split using prior knowledge of the IGHV mutational status evaluated, for example, by Sanger sequencing (Fig. 6d, e and Supplementary Data 2). Either approach successfully removed the nc-AID signature from U-CLL samples, in accordance with the pathogenesis of this CLL subgroup known not to be exposed to GC activity (Fig. 6d, e)[58].

This kind of a priori biological and clinical knowledge is not available for all cancer types. However, a simple clustering analysis based on the relative contribution of NNMF-extracted mutational signatures may also highlight the heterogeneity in signature activity and therefore help in the identification of distinct groups of patients, based on exposure to different mutational processes (Supplementary Fig. 11). Next, either a second NNMF run or a fitting approach using the NNMF shortlist can be performed on each single subgroup, as explained above[21].

This inter-sample bleeding of signatures is of course a universal phenomenon and as such can be also observed in non-B-cell hematological malignancies. To extend the validity of our findings we therefore focused on acute myeloid leukemia (AML), where we (i) performed WGS on two cases of therapy-related AMLs (t-AML) arisen after platinum-based chemotherapy for ovarian carcinoma and (ii) analyzed publicly available WGS data from the TCGA repository of primary AML cases (n = 50)[59]. In this setting, we extracted four main mutational processes: Signature 1, Signature 5 and two signatures currently not included in COSMIC. Of these, one was recently associated with platinum exposure (platinum signature) and the second to the

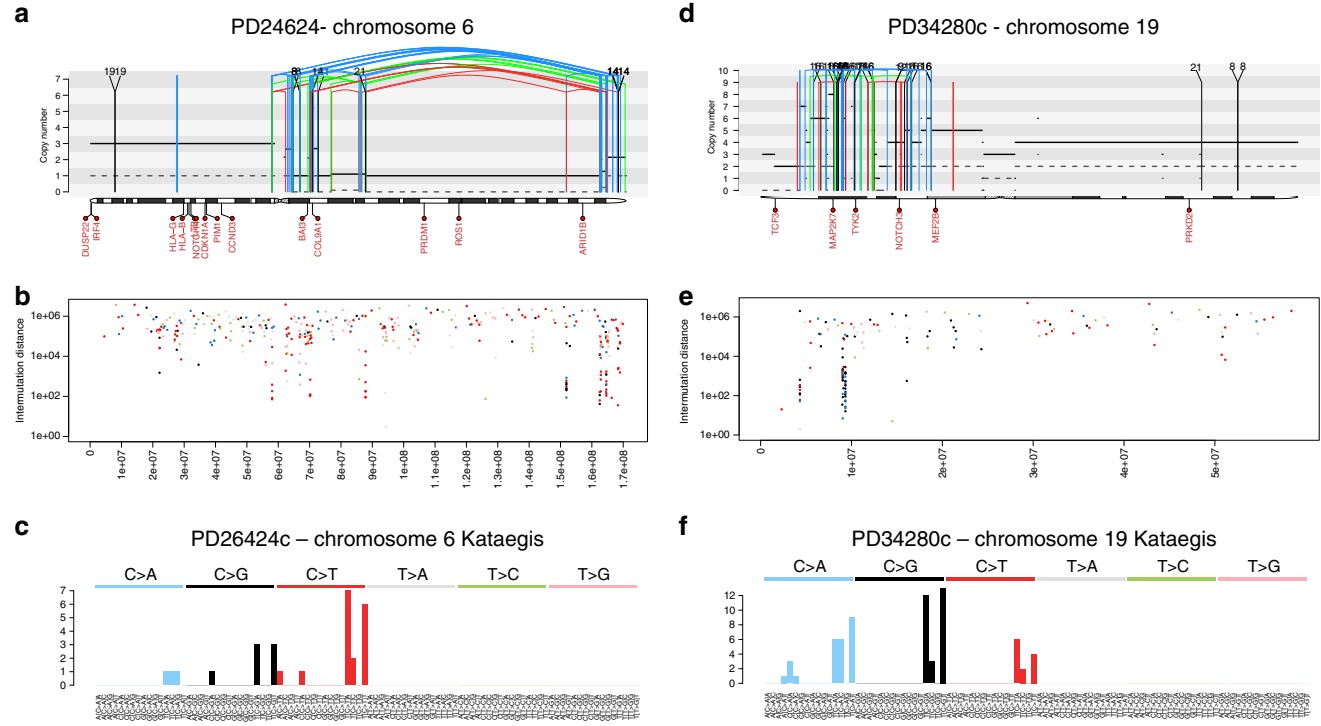

**Fig. 5** Kataegis in hematological malignancies. **a** Example of a MM patient with a chromothripsis on chromosome 6 associated with APOBEC-mediated kataegis. The solid and dashed lines reflect the total ploidy and the copy number status of the minor allele, respectively. In these plots, the red arch represents a deletion, the green arch represents a tandem duplication and the blue arch represents an inversion. **b** Inter-mutational distance of all mutations in chromosome 6, color-coded by mutational class. **c** Ninety-six-mutational classes of all kataegis events on chromosome 6. **d** Chromothripsis event on chromosome 19 in a therapy-related AML. **e** Inter-mutational distance of all mutations across chromosome 19. **f** Ninety-six-mutational classes of all mutations involved in the chromosome 19 kataegis: APOBEC emerged as the dominant mutational process, despite its activity was not detectable across the genome (Supplementary Software File 3)

hemopoietic stem cell nature (HSPC Signature) (Fig. 7a, b and Supplementary Data 1)[31,38,60–62]. The platinum signature contributed for >30% of the mutational burden of t-AMLs, but its activity was also found among primary AML from TCGA (Fig. 7c). This is inconsistent with the prior knowledge of these samples being treatment-naive. Confirming that platinum signature in primary AML samples represents a further example of inter-sample bleeding, analysis of TCGA primary AMLs without the two t-AML cases led to disappearance of the Platinum Signature (Fig. 7d and Supplementary Sofware 3). Furthermore, our analysis confirmed the added benefit of performing a de novo signature extraction as a first approach, as two out of four mutational signatures extracted in this cohort of 52 AMLs are not currently included in COSMIC.

## Discussion

In this study, we explored caveats and pitfalls of mutational signature analysis using whole-genome sequencing data from three common hematological neoplasms, focusing on the sample set preparation and post-algorithm interpretation processes. Furthermore, we showed how a comprehensive and detailed mutational signature analysis can provide relevant biological insights within different and well characterized cancer types, such as the c-AID activity among UM-IGHV, the absence of BRCA1/BRCA2-mediated HRD in a MM cohort and two mutational processes in AML, one related to platinum and one less characterized related to stem and progenitor bone marrow cells[31,38,60–62].

With the rapid increase in the number of tumor genomes sequenced, novel mutational signatures can be identified using several approaches discussed in this work. However, blind trust on out-of-the-box results from public tools can produce an incomplete representation of signatures, or the inclusion of false positives. Our results contain useful practical considerations that can resolve some of the uncertainty in the use of different algorithms, and in the interpretation of the results.

Important caveats and pitfalls a scientist can face in mutational signature analysis can usually be recognized and corrected by a priori knowledge of the biology of the tumor and by deep understanding of the way each algorithm works. For example, in CLL it is known that nc-AID exposure within the germinal center is only present among M-CLL cases. Therefore, the finding of Signature 9 activity in U-CLL must be regarded to as artefactual, related to the bleeding phenomenon that is common among de novo NNMF-based approaches. Knowing weaknesses and strengths of each approach, we proposed solutions to improve the accuracy of signature identification, with results that are biologically plausible. The main point of this study is in fact to highlight how the statistical and mathematical methods are important, but they must be used with expertize and combined with a good knowledge of the cancer type being studied. This is especially true when it comes to assignment of flat signatures: our original analysis demonstrates that the previously identified presence of BRCA1/BRCA2-like HRD in an MM cohort is likely to be a false-positive call of fitting algorithms[32], but this can only be demonstrated knowing the actual genomic consequences of BRCA deficiency in cancers and comparing them to what is seen in MM. Of course, our results only argue against the presence of BRCA1/BRCA2-type of HRD in our MM cohort, as we and others have convincingly demonstrated that a subset of MM patients are characterized by a significant grade of genomic instability[3,21,22,44,63–65].

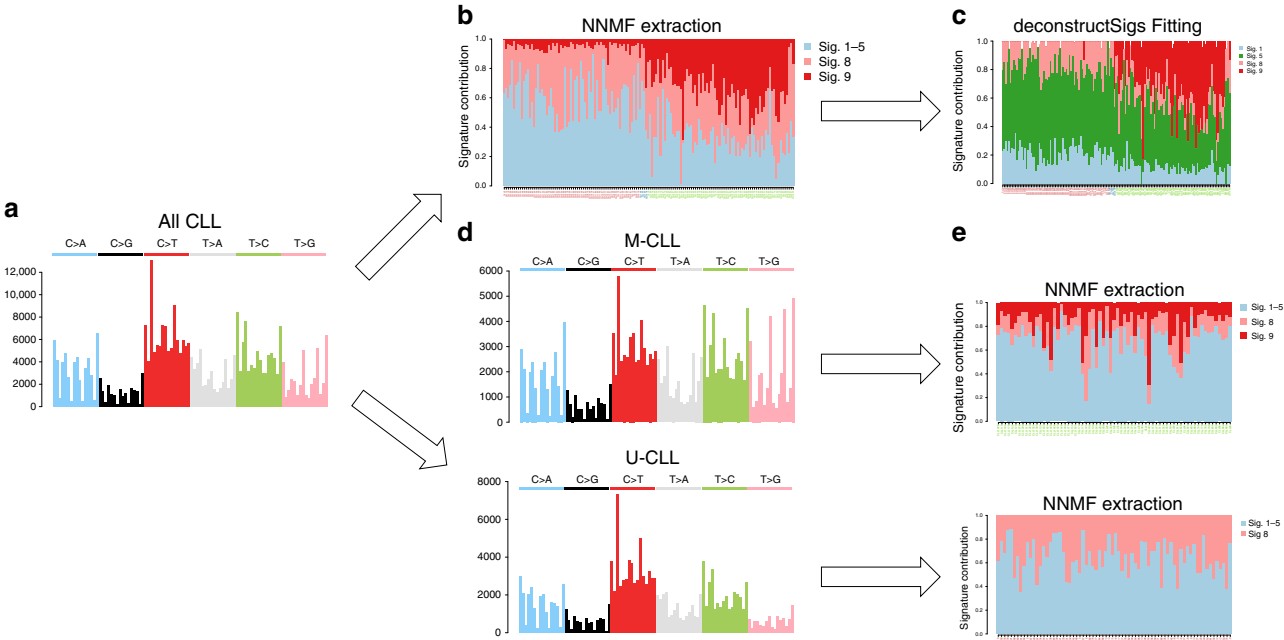

**Fig. 6** Bleeding of signatures in CLLs. Summary of mutational signature analysis on 146 CLL cases. From the 96-mutational catalog (**a**) the Alexandrov et al.[6,7] framework (NNMF) extracted different mutational processes. Signature 9 (nc-AID) was extracted also among U-CLL in contrast with their known pathogenesis (**b**). This is a typical example of inter-sample bleeding and it can be solved either running a fitting approach after the initial NNMF analysis using only the catalog of signatures extracted by NNMF (**c**), or analyzing M-CLL and U-CLLs in two different and independent runs (**d**, **e**). Using the 30 COSMIC signatures as reference, the first approach is usually the most appropriate in order to estimate the real contribution of each single mutational process. In fact, the NNMF extracted signatures may be over or under split, therefore preventing a precise estimation of their contribution. For example, in this analysis, Signature 1 and 5 were extracted as one single process and only by running a fitting approach we were able to differentiate these two processes (**c**). Sig. = signature. In **b** and **c**, red patient labels are used for U-CLL, green for M-CLL, and blue for unknown cases

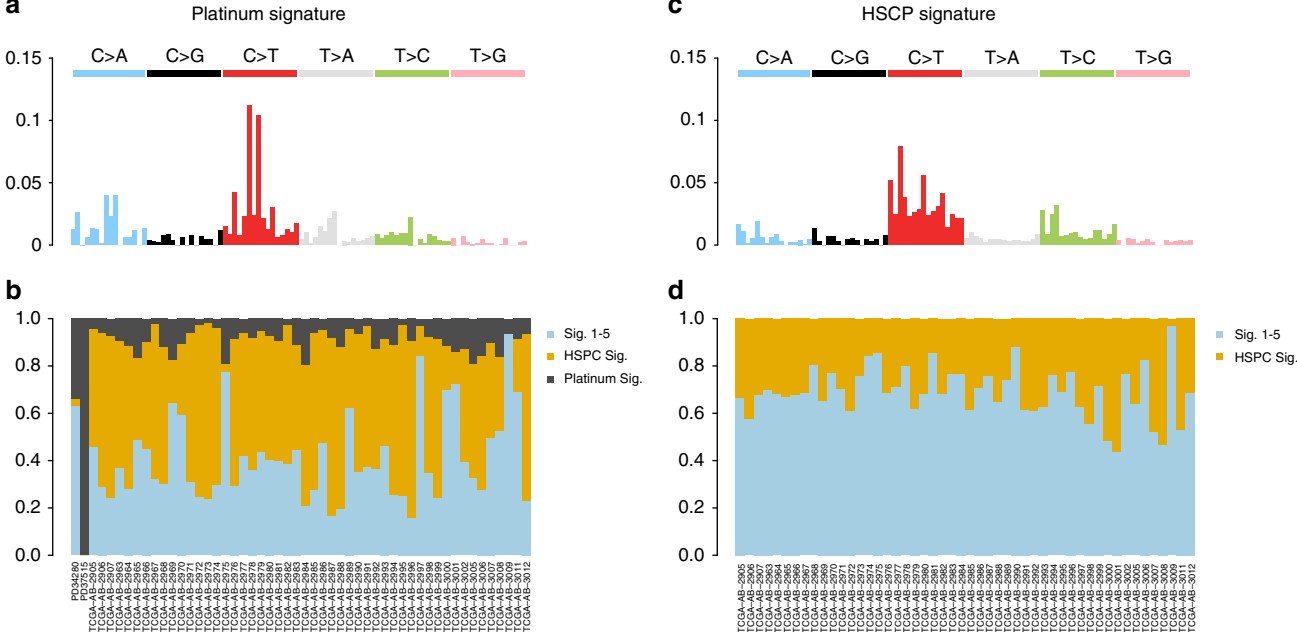

**Fig. 7** Bleeding of signatures in AMLs. Example of inter-sample bleeding among 52 AML WGSs. **a**, **b** Running NNMF on the entire cohort, we extracted two mutational signatures not currently included in COSMIC: one recently associated with platinum exposure and the second recently reported as a process specific to the hemopoietic stem cell (HPSC). **c**, **d** The inclusion of two t-AMLs (PD34280 and PD37515) affects the global signature extraction, with Platinum Signature extracted also in the primary AMLs. Removing the t-AMLs the inter-sample bleeding was corrected, and no Platinum Signature was extracted in primary AMLs. Sig. = signature

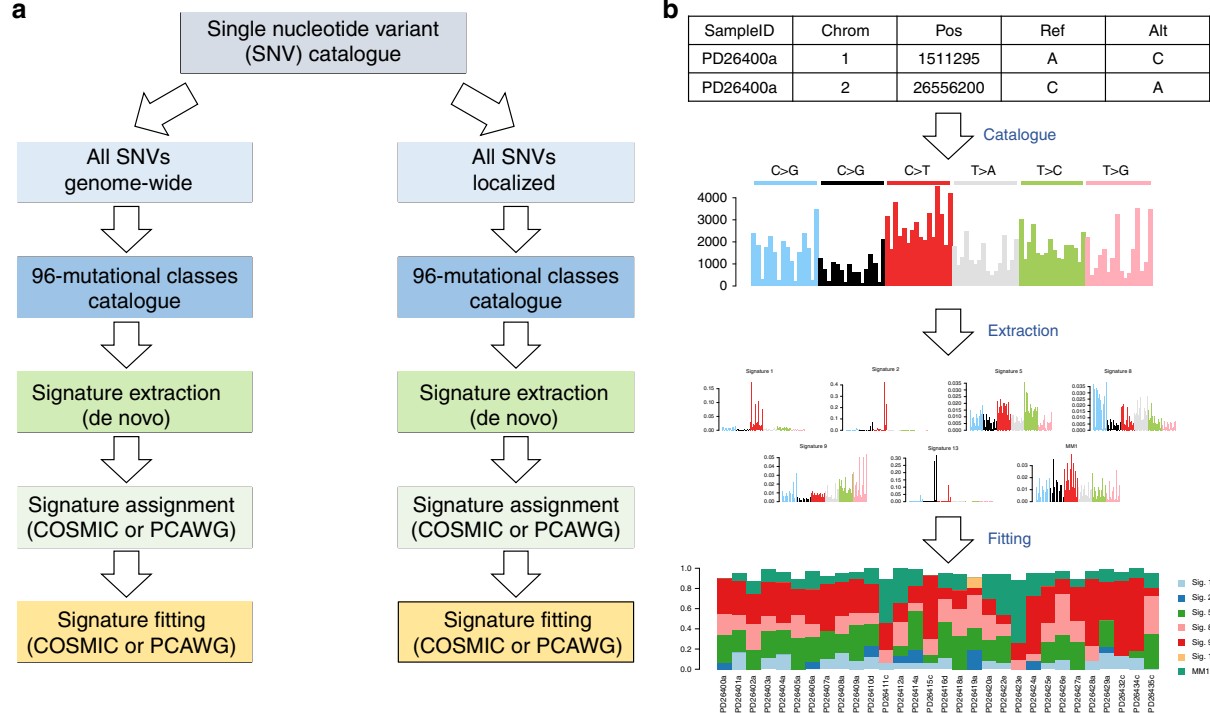

**Fig. 8** Mutational Signature workflow. Our suggested workflow for mutational signature analysis for both genome-wide and clustered processes (**a**) and an example of its application on 30 MM WGSs (**b**)

In general, our preferred approach to investigate mutational signatures in hematological malignancies follows three different steps: (1) signature discovery with a de novo extraction process; (2) assignment of extracted signatures to a reference catalog (i.e., COSMIC) and possibly identification of novel ones; (3) a fitting approach including only the subset of COSMIC signatures identified from the extraction process (Fig. 8). This multi-step approach allows the identification of known and novel signatures and their correct quantification, avoiding artefactual calls related to bleeding and overfitting. Based on a similar approach, two novel robust and stringent tools have recently been developed allowing the identification of >30 new mutational signatures and the redefinition of the previous 30-COSMIC signatures, creating a catalog to be used as reference for future studies[31]. These improved knowledge banks and bio-informatic tools will further refine our ability to investigate mutational signatures in hematological malignancies. However, we are convinced that prior knowledge of cancer biology and genomics will always be indispensable for a correct data interpretation.

## Methods

**Sample selection and processing of genomic data**. In this study, we analyzed the single-nucleotide variant (SNV) catalog from four WGS cohorts: 143 CLLs (EGAS00000000092)[52,53], 30 MMs (EGAD00001003309)[3,9], 50 AMLs (phs000178.v1.p1)[59], and two unpublished t-AML (EGAD00001005028). These last two cases were sequenced after written informed consent was obtained at the Wellcome Sanger Institute using the X10 Illumina platform. FASTQ files were aligned to the reference genome using BWAmem, and deduplicated aligned BAM files were analyzed using the following tools: ASCAT for copy number changes, BRASS for structural variations (large inversions and deletions, translocations, internal tandem duplication), Caveman and Pindel for Single-Nucleotide Variants (SNVs) and small insertion-deletions[20,66–68], respectively. The characterization of the main clinical and genomic features of MM and CLL series is summarized in Supplementary Table 1 and Supplementary Data 2, respectively. Kataegis was defined as a cluster of 6 or more consecutive mutations with an average intermutation distance of less than or equal to 1 Kb[20].

The study involved the use of human samples, which were collected after written informed consent was obtained (Wellcome Trust Sanger Institute protocol

number 15/046 for the myeloma samples, Fondazione IRCCS Istituto Nazionale dei Tumori code 127/16 for the t-AML samples).

**Mutational signature workflow**. Mutational signatures were investigated using three published and available algorithms: the Alexandrov et al.[6] NNMF framework, deconstructSigs[24] and mutationalPatterns[37] R packages. The full mutationalPatterns analysis was written in R and the code is provided in Supplementary Software Files 1–3 for MM, CLL, and AML respectively. Each of the above methods produces a matrix decomposition $C \approx SE$, where $C$ is the catalog matrix, with mutation types as rows and samples as columns, $S$ is the signature matrix, with mutation types as rows and signatures as columns, and $E$ is the exposure matrix, with signatures as rows and samples as columns (Supplementary Fig. 1). The reconstruction error indicates how similar the mutational profiles of samples in $C$ are to those in the product $SE$, and can be computed using different metrics, such as cosine similarity, Kullback-Leibler divergence (KLD) or RMSE.

Each of the signatures extracted with either mutationalPatterns or the method from Alexandrov et al.[6,7,37] were assigned to one or a combination of two COSMIC signatures. To do so, cosine similarities between the extracted signatures and each COSMIC signature, or a linear combination of two COSMIC signatures (using non-negative least squares R package NNLS), were computed. These results are available in Supplementary Data 1.

**HRDetect in multiple myeloma**. Analysis of homologous recombination deficiency (HRD) from BRCA1/BRCA2 deficiency as a possible source of genomic instability was performed using the recently published HRDetect algorithm[18]. The structural variant and indel catalog in MM were generated using BRASS and Pindel, respectively[20,67].

**Single-nucleotide variants on IGH**. The mutation cancer cell fraction for c-AID SNVs was estimated using the Dirichlet process for both CLLs and MMs[4,9]. Considering the well-known complexity and low-quality mappping of IGH region, we ran three additional SNV callers (mutect2[69], caveman[66], and muse[70]) to reduce the rate of false positives and we combined the results with the published catalog of SNVs generated with Sidron[52]. Seventy-nine percent of the previously published mutations on IGH was confirmed by at least one additional caller (Supplementary Fig. 12). Furthermore 512 additional SNVs were called by at least two out of the three new callers. Only mutations called by at least 2 out of 4 callers were included in the final analysis.

**Reporting summary**. Further information on research design is available in the Nature Research Reporting Summary linked to this article.

## Code availability

All R codes used to generate signature data using mutationalPatterns in the paper are provided as Supplementary Software Files 1–3. All codes have been generated using R software v. 3.4.2.

## Data availability

The sequencing data pertaining to MM are available from the European Genome-phenome archive (EGA) database under the accession code EGAD00001003309. The sequencing data pertaining to CLL are available from EGA under the accession code EGAS00000000092. The published and unpublished AML sequencing data are available from dbGAP under the accession code phs000178 and from EGA dbGAP under the accession code EGAD00001005028, respectively. The breast cancer WGSs are available from the EGA under the accession code EGAS00001001178[20]. All the other data supporting the findings of this study are available within the article and its supplementary information files and from the corresponding author upon reasonable request. A reporting summary for this article is available as a Supplementary Information file.

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

## Acknowledgements

F.M. is supported by A.I.L. (Associazione Italiana Contro le Leucemie-Linfomi e Mieloma ONLUS), by S.I.E.S. (Società Italiana di Ematologia Sperimentale) and by the Memorial Sloan Kettering Cancer Center NCI Core Grant (P30 CA 008748). N.B. is funded by the University of Milan (project 22597-PSR2017_DIP_032) and by the European Research Council under the European Union's Horizon 2020 research and innovation program (grant agreement no. 817997). X.S.P. is supported by thr Ministerio de Economía y Competitividad Grant No. SAF2017–87811-R. F.N. is supported by a pre-doctoral fellowship of the MINECO (BES-2016–076372). This work was supported by the Instituto de Salud Carlos III (project PMP15/00007, F.N., E.C.), the "la Caixa" Foundation Grant No HR17-00221 (Health Research 2017 Program, F.N., E.C.), the Ministerio de Economía y Competitividad (MINECO) SAF2013-45836-R (E.C.) from. Plan Nacional de I + D + I, Generalitat de Catalunya Suport Grups de Recerca AGAUR 2017-SGR-1142 (E.C.) and the European Regional Development Fund "Una manera de hacer Europa". E.C. is supported by ICREA under the ICREA Academia program. A.D. is funded by a CRUK Pioneer Award C60100/A23433. S.N.Z. is funded by a CRUK Advanced Clinician Scientist Award (C60100/A23916) and a CRUK Grand Challenge Award (C60100/A25274). This work was supported by: Department of Veterans Affairs Merit Review Award I01BX001584-01 (N.C.M.), NIH grants P01-155258 (N.C.M., H.A. L., M.F., P.J.C., K.C.A.) and 5P50CA100707-13 (N.C.M., H.A.L., K.C.A.). We thank Michael R. Stratton for discussions and help in data interpretation.

## Author contributions

F.M. designed the study, collected, and analyzed the data and wrote the paper; N.B. designed the study, collected the data, and wrote the paper; A.D. analyzed the data and wrote the paper, H.D., D.L., L.M., F.N., B.Z. and R.R. analyzed the data; H.A.L., X.P., E.C., P.J.C., S.N. and N.M. collected the data.

## Additional information

**Competing interests:** The authors declare no competing interests.

