## [Peer Review File · Nature Communications]

Reviewers' comments:

Reviewer #1 (Remarks to the Author):

The authors look into various issues relevant to the accuracy in mutational motif prediction, considering CLL and MM cancer cases, and propose a general methodology for future analyses.

1. It was unclear why COSMIC signatures 3, 5 and 8 in particular were chosen as examples of "flat" distributions, and for the analysis throughout. The choice seems arbitrary so there needs to be some objective procedure shown for having chosen these three.
2. Line 167 and below: Need explanation of how de novo signatures were determined to be "merged" versions of COSMIC signatures.
3. The Ig loci mutations of Supp Fig 7 show a large concentration of mutations in what are presumably rearranged VDJ regions. Although the issue of dealing with rearrangement is alluded to (Fig S8), more details should be given for what was done here to ensure these are real mutations.
4. There are problems with the quality of the English from the beginning and throughout, e.g. line 70, "Despite the great majority ...".
5. Explanation around C=SE starting on line 113 is not informative, starting with the fact that C, S, E are not even defined (although this is done in the Methods). A diagram would be helpful.
6. Many of the figures contain text that is so small as to be useless, e.g. Fig 3b.
7. In Fig 5e – are the "de novo" mutational signatures being labeled as "Sig 9" here? It isn't clear.

Reviewer #2 (Remarks to the Author):

The authors have compared the pros and cons of public signature analysis tools using available WGS data generated from CLL and multiple myeloma patients. They focus on three aspects; mutational signature analysis using a de novo extraction vs a fitting approach, the issue of localized hypermutation, e.g. caused by AID, and finally, inter-sample bleeding. They reveal potential erroneous signature assignment, as exemplified by signature 3 in multiple myeloma, and propose an analysis framework for future, reproducible mutational signature detection.

Major comments:

The authors claim that they provide new insights by demonstrating evidence of c-AID activity among unmutated CLL cases. This is not novel and the data provided does not contribute significantly to our current knowledge.

The authors have selected two B-cell malignancies that both originate from B-cells that have undergone somatic hypermutation. Hence, since this is a phenomenon specific to B-cell derived neoplasms, the data on localized hypermutation is less relevant for other hematological malignancies, let alone other cancer types.

The analysis of inter-sample bleeding is based on two clinical groups of CLL with distinct somatic hypermutation status and the data demonstrated is indeed expected. Other examples how their approach could reveal inter-sample bleeding should be provided.

In this Reviewer's perspective, they should have extended their analysis beyond B-cell malignancies, i.e. to other cancer types, to show the reproducibility of their proposed analysis strategy.

Below, we include a point-by-point response to the reviewer comments – the reviewer comments are in black, our response in blue and the actions we have taken in red.

Reviewers' comments:

Reviewer #1 (Remarks to the Author):

The authors look into various issues relevant to the accuracy in mutational motif prediction, considering CLL and MM cancer cases, and propose a general methodology for future analyses.

1. It was unclear why COSMIC signatures 3,5 and 8 in particular were chosen as examples of “flat” distributions, and for the analysis throughout. The choice seems arbitrary so there needs to be some objective procedure shown for having chosen these three.

We agree with the reviewer that this part of the manuscript was not sufficiently explained in the original manuscript. In fact, previous literature has pointed at these signatures as ones that can explain equally well different mutational catalogues and pose problems with assignment due to the lack of distinctive mutational peaks -i.e., their “flat” distribution- (see for example Huang, X. et. All Bioinformatics 2017). Furthermore, these 3 signatures are the 3 main ones reported across different haematological malignancies (Alexandrov et al. Nature 2013 and BioRxiv 2018), and thus of particular relevance for the whole analysis. This is now better described in the revised version of the manuscript and supplementary material.

2. Line 167 and below: Need explanation of how de novo signatures were determined to be “merged” versions of COSMIC signatures.

In the methods section of the revised version of the text we have better explained the methodological process behind the de novo mutational signature extraction and their assignment. This was based on a cosine similarities approach between the extracted signatures and each COSMIC signature, or a linear combination of two COSMIC signatures in case of assignment to “merged” signatures (using non-negative least squares R package NNLS). The results of these computations are available in the new Supplementary Table 2 of the revised manuscript so that the reader can better understand the process and reproducibility of the analysis is facilitated.

3. The Ig loci mutations of Supp Fig 7 show a large concentration of mutations in what are presumably rearranged VDJ regions. Although the issue of dealing with rearrangement is alluded to (Fig S8), more details should be given for what was done here to ensure these are real mutations.

Indeed the VDJ and class switch recombination loci on the IGH region are known to be affected by low mapping quality due to the presence of repeat regions and several structural variations. These particular features can affect our specificity and sensitivity in calling single nucleotide variants. Overall, the CLL IGH mutational landscape is in line with the IGHV mutational status evaluated by the standard sanger sequencing (Figure 4d), suggesting a good quality of the published calls. However, to further improve the accuracy of these catalogue of mutations on the IGH we have run 3 different SNV callers (Mutect, Muse and Caveman) and we combined the results with the publish catalogue of SNVs. Seventy-nine percent of the previously published mutations on IGH was confirmed by at least one additional caller. The new consensus catalogue generated by this multi-pipeline approach confirmed the IGHV mutational status in all cases with available VDJ sanger sequencing data. Overall, these data confirm the good quality and efficiency of our bio-informatic tools to detect mutations on the IGH locus. The consensus catalogue for SNVs on IGH have been used to regenerate the new Figure 4 and Supplementary Figure 8. The concordance rate between different pipelines has been included in the new Supplementary Figure 12.

4. There are problems with the quality of the English from the beginning and throughout, e.g. line 70, "Despite the great majority ...".

English has been thoroughly revised and changed in the new manuscript.

5. Explanation around C=SE starting on line 113 is not informative, starting with the fact that C,S,E are not even defined (although this is done in the Methods). A diagram would be helpful.

To facilitate understanding of the methods used, in the revised version of the paper we expanded the explanation of the formula underlying the decomposition matrix right in the result section. Furthermore, as suggested by the reviewer, we included a diagram to better explain this formula as supplementary figure 1.

6. Many of the figures contain text that is so small as to be useless, e.g. Fig 3b.

We agree with the reviewer that in many parts of different figures labels were too small. Therefore, in the revised version of the paper we increased the size of labels and ensured these are legible in all figures.

7. In Fig 5e – are the "de novo" mutational signatures being labeled as "Sig 9" here? It isn't clear.

We agree with the reviewer that the term "de novo" in the label may misguide in the interpretation of the figures. To avoid any misunderstanding, we removed "de novo" from all Figure 5 (now Figure 6 of the revised paper). Furthermore, since sig. 9 is not extracted in U-CLLs, it was removed from the Figure legend.

Reviewer #2 (Remarks to the Author):

The authors have compared the pros and cons of public signature analysis tools using available WGS data generated from CLL and multiple myeloma patients. They focus on three aspects; mutational signature analysis using a de novo extraction vs a fitting approach, the issue of localized hypermutation, e.g. caused by AID, and finally, inter-sample bleeding. They reveal potential erroneous signature assignment, as exemplified by signature 3 in multiple myeloma, and propose an analysis framework for future, reproducible mutational signature detection.

Major comments:

The authors claim that they provide new insights by demonstrating evidence of c-AID activity among unmutated CLL cases. This is not novel and the data provided does not contribute significantly to our current knowledge.

We agree with the reviewer that the canonical AID activity in CLL was showed many years ago before the next generation sequencing era. In term of signatures, Kasar et al clearly showed that this mutational process is active also in UM-IGHV and also in the subclonal variants suggesting an ongoing activity. As a matter of the fact all these papers and data were reported in our explanation. However, we don't recall any data regarding the focal c-AID mutational activity on the class switch recombination, which represent the only claimed novelty in this specific part of the manuscript. Furthermore, we believe that our data provide the first explanation why VDJ coding part is unmutated in sanger sequencing despite a high c-AID activity on the CSR.

The authors have selected two B-cell malignancies that both originate from B-cells that have undergone somatic hypermutation. Hence, since this is a phenomenon specific to B-cell derived neoplasms, the data on localized hypermutation is less relevant for other hematological malignancies, let alone other cancer types.

Of course, somatic hypermutation promoted by AID is a distinct feature of B-cell malignancies. Nevertheless, it is well known that other mutational processes can act in a similar way across the genome. These localized hypermutation processes are usually called kataegis and have been detected in several B and non-B malignancies. We agree with the reviewer in that extending this analysis to other malignancies and to other mutational processes would strengthen the relevance of our paper. To this end, in the revised version of the manuscript we performed novel analysis on additional cancer samples to include instances of localized hypermutation promoted by APOBEC in MM and importantly in AML, where this phenomenon was never reported before. **These new results are reported in the new figure 5 of the revised manuscript and in the text (line 351-362).**

The analysis of inter-sample bleeding is based on two clinical groups of CLL with distinct somatic hypermutation status and the data demonstrated is indeed expected. Other examples how their approach could reveal inter-sample bleeding should be provided. In this Reviewer's perspective, they should have extended their analysis

beyond B-cell malignancies, i.e. to other cancer types, to show the reproducibility of their proposed analysis strategy.

We agree with the reviewer that the inclusion of a non-B-cell cancer type would strengthen the message of our paper. Therefore, in the revised version of the paper we performed additional analysis on WGS data from 50 primary acute myeloid leukemias (AMLs) published in NEJM in 2012. Furthermore, we also purposely sequenced the whole genome of 2 cases of therapy-related AMLs. With this additional data, we provide evidence of the relevance of our analysis beyond B-cell malignancies. Specifically, we report on additional examples of localized hypermutation (see above) and of inter-sample bleeding between primary and therapy-related AML samples. Data from this new analysis is now part of the revised manuscript in the form of 2 additional figures (Figure 5 and 7, respectively) and is commented on the text (lines 397-416).

REVIEWERS' COMMENTS:

Reviewer #1 (Remarks to the Author):

I'm satisfied with the revisions made.

Reviewer #2 (Remarks to the Author):

The authors have addressed all major concerns and the manuscript has been significantly improved, in particular by addition of new data on AML.